

# Entropy of causal diamond ensembles

**Ted Jacobson[1,2]⋆ and Manus R. Visser[2]†**

**1** Maryland Center for Fundamental Physics, University of Maryland,
College Park, MD 20742, USA
**2** Department of Applied Mathematics and Theoretical Physics, University of Cambridge,
Wilberforce Road, Cambridge CB3 0WA, UK

⋆ jacobson@umd.edu , † mv551@cam.ac.uk

## Abstract

We define a canonical ensemble for a gravitational causal diamond by introducing an artificial York boundary inside the diamond with a fixed induced metric and temperature, and evaluate the partition function using a saddle point approximation. For Einstein gravity with zero cosmological constant there is no exact saddle with a horizon, however the portion of the Euclidean diamond enclosed by the boundary arises as an approximate saddle in the high-temperature regime, in which the saddle horizon approaches the boundary. This high-temperature partition function provides a statistical interpretation of the recent calculation of Banks, Draper and Farkas, in which the entropy of causal diamonds is recovered from a boundary term in the on-shell Euclidean action. In contrast, with a positive cosmological constant, as well as in Jackiw-Teitelboim gravity with or without a cosmological constant, an exact saddle exists with a finite boundary temperature, but in these cases the causal diamond is determined by the saddle rather than being selected a priori.



# 1 Introduction

Since the Bekenstein-Hawking entropy [1,2] of a black hole scales locally with the area of the horizon, it seems that the presence of a horizon itself entails entropy, regardless of the global structure of the horizon [3,4]. This is certainly consistent with black hole entropy, the entropy of acceleration horizons [5], and the entropy of entanglement wedges in the context of AdS/CFT holography [6–8]. Gibbons and Hawking (GH) derived the entropy of black hole and de Sitter horizons from a Euclidean saddle approximation of the gravitational partition function [9], and a similar method has been applied for entanglement wedges [10–12]. The case of a static patch in de Sitter spacetime differs from the black hole and entanglement wedge cases in that there is no boundary on which to anchor the specification of the states being considered. Nevertheless, as GH found, the saddle action yields the expected entropy. Moreover, one can introduce an artificial boundary inside the de Sitter horizon at which to define ensemble state parameters [13–21]. A static patch in de Sitter spacetime being a particular case of a causal diamond with an edge of finite area, it is natural to ask whether also the entropy of a causal diamond [22–29] in Minkowski spacetime—or in any maximally symmetric spacetime—can be computed by the GH method or something similar.

This question was addressed in a recent paper by Banks, Draper and Farkas (BDF) [30], by evaluating the action for a Euclidean analytic continuation of the diamond metric. Using this approach, which parallels well established computations of Killing horizon entropy, they obtain the expected result $A/4$, where $A$ is the area (in Planck units[1]) of the edge of the diamond. While evidently correct on some level, the fundamental basis of this computation remains obscure. What is lacking, from our viewpoint, is a conceptual framework in which the computation yields an approximation to the entropy of a well-defined ensemble. In this note we provide such a framework.

We begin with a brief summary of the computation in [30].[2] A causal diamond in a maximally symmetric spacetime admits a conformal Killing vector $\zeta$ whose flow preserves the diamond, and the diamond horizon is a conformal Killing horizon with respect to $\zeta$ [29]. Although $\zeta$ is not a true Killing vector, and thus the diamond is not an "equilibrium" configuration in the usual sense, it is an "instantaneous" Killing vector on the maximal slice of the diamond, in particular at the edge where the Killing vector vanishes. The diamond admits a natural conformal Killing time coordinate $s$ such that $s = 0$ on the maximal volume slice of the diamond and $\zeta \cdot ds = 1$. BDF analytically continue $s$ to imaginary values $s = -is_{\mathrm{E}}$, and periodically identify $s_{\mathrm{E}}$ with $s_{\mathrm{E}} + 2\pi$.[3] For a $D$-dimensional Minkowski diamond this results in a flat Euclidean spacetime in which the fixed point set of the analytically continued conformal Killing vector, i.e., the Euclidean horizon, is a surface of topology $S^{D-2}$ (see Figure 1). We shall refer to this analytic continuation of the diamond as the "Euclidean diamond".

BDF adopt the viewpoint that the horizon should be excluded from the Euclidean domain, and introduce an infinitesimal boundary around it and a Gibbons-Hawking-York (GHY) boundary term in the gravitational action that evaluates to $-A/4$ (in agreement with the well-known result for black hole horizons [31]). They interpret the value of the Euclidean action as minus the entropy, on the grounds that the period of the infinitesimal time circle at the boundary is zero, corresponding to infinite temperature, so the action—which for gravitational partition function is the free energy $U - TS$ divided by the temperature $T$—reduces to $-S$.

The answer is satisfactory, but what are the principles behind the calculation? In particular, why is the horizon excluded by a boundary, what determines the added boundary term, and

---

[1]In this paper we adopt Planck units, $\hbar = c = G = 1$, and mostly plus spacetime signature, with dimension $D$ unless otherwise specified.

[2]Ref. [30] studies causal diamonds in maximally symmetric spacetimes, as well as in Schwarzschild and Schwarzschild-de Sitter spacetime. Here we restrict attention to the former cases.

[3]The norm of $\zeta$ is chosen such that the surface gravity of the horizon is equal to one.

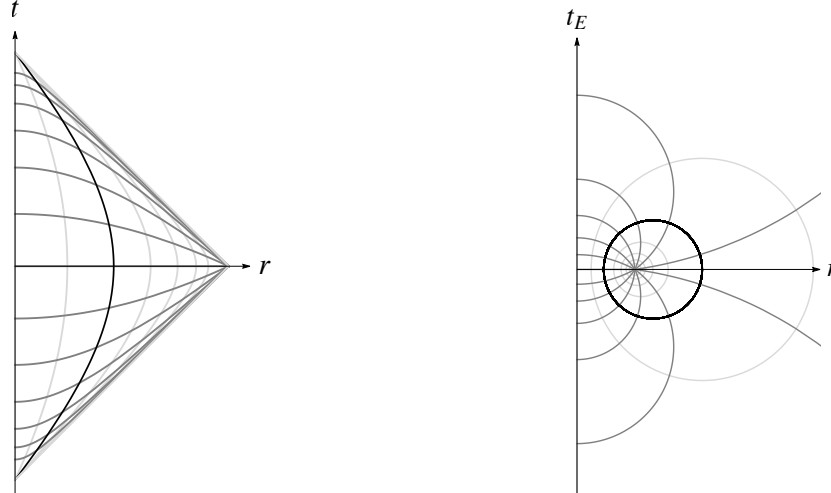

Figure 1: Minkowski causal diamond (left) and its Euclidean continuation (right) where an $S^{D-2}$ is suppressed. Orbits of the conformal Killing vector $\zeta$ (light gray curves) are uniformly accelerated in the Minkowski diamond and circular in the Euclidean one. Curves of constant conformal Killing time $s \in (-\infty, \infty)$ (gray), defined such that $s = 0$ on the maximal slice and $\zeta \cdot ds = 1$, and curves of constant radial coordinate $x \in [0, \infty)$ (light gray), satisfying $\zeta \cdot dx = 0$ and $|dx| = |ds|$, are plotted at equal coordinate intervals of 0.5. In the right figure the Euclidean horizon is the fixed point set of the analytically continued conformal Killing vector, which is located at $x = \infty$. In both figures a conformally stationary York boundary is shown in black at $x = 1$ that becomes stationary in the near-horizon limit.

why is the absence of a true Killing vector not problematic for an equilibrium state? To clarify the significance of the BDF calculation we need to make more precise the question it purports to answer. That is, we need to identify the *ensemble* whose entropy is being computed.

## 2 Causal diamond ensembles with a York boundary

Rather than starting with a particular solution to Einstein's equation and analytically continuing to Euclidean signature, in principle one should start with a partition function for a gravitational ensemble and find its saddle point approximation. To this end, we will follow the method pioneered by York [32] for black hole ensembles, which has also been applied to ensembles with cosmological horizons [13–21]. The idea is that if the physical context does not supply a natural boundary at which the ensemble is specified, one can introduce an artificial boundary with data that do so.

### 2.1 Canonical ensemble

A canonical ensemble is defined relative to a "York boundary" equipped with a time flow vector field and some choice of fixed conservative boundary conditions, including a specified period in imaginary time, i.e., the inverse temperature. In the present context we are interested in saddle configurations in which the boundary sits *inside* the diamond, in the sense that its area is smaller than that of the surrounding spheres in the system. For simplicity we adopt here Dirichlet boundary conditions by fixing the induced boundary metric, which specifies both the

spatial geometry of the boundary and the period in imaginary time, despite the fact that for such boundary conditions the initial boundary value problem is likely not well posed [33]. Whether the conclusions of our analysis regarding the saddles and their actions would be similar for, e.g., the conformal boundary conditions discussed in [33–36]—which seems likely to be the case—is a question worthy of investigation.

The canonical ensemble is then defined by the density matrix

$$\rho = \frac{1}{Z} \exp(-\beta H_{\mathrm{BY}}), \tag{1}$$

where $\beta$ is the inverse proper temperature that is held fixed at the boundary, $H_{\mathrm{BY}}$ is the Brown-York (BY) Hamiltonian [37], and $Z = \mathrm{Tr}\exp(-\beta H_{\mathrm{BY}})$ is the canonical partition function. On the constrained phase space $H_{\mathrm{BY}}$ is given by

$$H_{\mathrm{BY}} = -\frac{1}{8\pi} \oint d^{D-2}x \sqrt{\sigma} k, \tag{2}$$

where the integral is over a spatial cross-section of the boundary with metric $\sigma$, and $k$ is the trace of the extrinsic curvature of the boundary as embedded in a spatial slice of the bulk, defined with respect to the outward normal from the system. $H_{\mathrm{BY}}$ generates the system's proper time evolution orthogonal to the spatial cross-sections of the boundary.

Of course the partition function for quantum gravity cannot be evaluated exactly, and general relativity (GR) is not a UV complete theory so an exact partition function in that theory alone does not exist in any case. Nevertheless, as first demonstrated in the pioneering paper of Gibbons and Hawking [9], apparently meaningful approximations can be obtained using a saddle point approximation and the action of the low energy effective theory. The canonical ensemble is in principle time independent, so the boundary conditions defining the ensemble should share that property. If a saddle exists, the spherically symmetric boundary $S^{D-2} \times S^1$, with a product metric and $S^1$ circumference $\beta$, should be embeddable in the saddle and generated by the flow of a spherically symmetric Killing vector field. One such saddle has topology $\mathrm{ball}^{D-1} \times S^1$, and corresponds to a solid cylinder in periodically identified Euclidean space, $\mathbb{R}^{D-1} \times S^1$. This saddle approximates the contribution of "hot flat space" to the partition function, and does not include the horizon, hence misses the horizon entropy.

Instead, we seek a saddle with topology $S^{D-2} \times D^2$, where the boundary of the disk $D^2$ is the thermal time circle at the York boundary, which contracts to a point at the center of the $D^2$ where the horizon lies. Furthermore, we require that the horizon area is larger than the boundary area, because we want to describe (Euclidean) causal diamonds (not black holes). The existence of such a saddle would require the existence of a Ricci-flat $D$-geometry in which $S^{D-2} \times S^1$ (with the product metric) could be embedded. We do not know whether such a saddle might exist for $D \geq 4$, but for $D = 3$ it clearly does not. In $D = 3$ dimensions, Ricci flat implies flat, and the boundary is an intrinsically flat torus $S^1 \times S^1$, which cannot be isometrically embedded in 3-dimensional Euclidean space $\mathbb{R}^3$. Moreover, in any dimension $D \geq 3$, we can see that (a portion of) the Euclidean diamond is *not* a saddle that meets the boundary conditions, since it admits no spherically symmetric Killing vector. On the other hand, the Euclidean diamond does admit a spherically symmetric *conformal* Killing vector, and one could be tempted to allow the ensemble boundary to coincide with one of its orbits, as illustrated by the black line in Figure 1. But this is unacceptable for a stationary ensemble since, as is clear from the figure, the radius of the boundary sphere is not constant along this orbit.[4] If at the Euclidean time $s_{\mathrm{E}} = 0$ the boundary is a $(D-2)$-sphere of radius $r_0$, concentric

---

[4]One could instead generalize the definition of partition function to allow for time dependence of the boundary conditions, namely those induced on a boundary following orbits of the conformal Killing field of the Euclidean diamond. Then a saddle would exist, but it would not be a saddle of a true thermal partition function.

with the horizon of radius $R$, then along the Killing orbit the radius of the sphere grows to a maximum value of $R^2/r_0$ at $s_E = \pi$,[5] corresponding to a sphere on a spatial slice outside the diamond in the Lorentzian geometry,[6] and then returns to $r_0$ at $s_E = 2\pi$. The sphere hence sweeps out a surface of topology $S^{D-2} \times S^1$ with different area of the $S^{D-2}$ at different points along the $S^1$. In other words, the area radius of the boundary depends on the conformal Killing time $s_E$.

Despite the absence of a true horizon saddle, all is not lost. The horizon sphere is fixed under the Euclidean conformal time flow, so as the orbit approaches the horizon, it becomes stationary, hence can be identified with the York boundary defining a bona fide canonical ensemble. In the near-horizon limit, the length of a Euclidean conformal time circle goes to zero, hence a Euclidean diamond can be a good approximation to a saddle for an ensemble with boundary radius $r_0$ and temperature $T$ provided that $T \gg 1/r_0$.[7] The horizon radius of that approximate saddle will approach the radius $r_0$ of the boundary defining the ensemble as $T$ goes to infinity.

## 2.2  Microcanonical ensemble

A microcanonical ensemble with fixed BY energy was defined for black holes in [38], and that construction has been extended to the case of cosmological horizons (for a recent review see [19]). In the present case, however, such an ensemble would admit no horizon saddle since, even in the near-horizon limit, the BY energy of the Euclidean diamond is not constant on a York boundary that follows a circular orbit of the Euclidean Killing flow.[8] This is clear from Figure 1: the BY energy is proportional to the trace of the spatial extrinsic curvature of the boundary sphere on a spatial slice, which is proportional to the rate of change of the sphere radius $r$ with respect to distance normal to the boundary along the spatial slice. The spatial slices are indicated by the grey curves in the figure. The angle at which these curves meet the York boundary (whose Euclidean "history" is indicated by the black circle) rotates through $2\pi$ around the boundary, hence the spatial derivative of $r$ oscillates, changing sign twice around the Euclidean conformal time circle, no matter how close to the horizon the York boundary lies. The BY energy therefore oscillates in Euclidean conformal time. Expressed differently, in diamond universe coordinates the trace of the extrinsic curvature of a $(D-2)$-sphere as embedded in a constant $s_E$ spatial slice is: $k = \frac{D-2}{R} \sin s_E$, which notably does not depend on $x$ but rather on the Euclidean time $s_E$, hence also at the Euclidean horizon $k$ is time dependent. Thus, since there is no horizon saddle for the microcanonical ensemble, we shall now focus exclusively on the canonical ensemble.

---

[5]The radius $r$ of the sphere on the circular closed orbits of $\zeta$ can be computed by expressing it in terms of the "diamond universe" coordinates $(s_E, x)$ that cover a causal diamond in Euclidean flat space: $r = R \sinh x/(\cos s_E + \cosh x)$ [30]. Multiplying $r_0 \equiv r(s_E = 0)$ and $r_\pi \equiv r(s_E = \pi)$ yields $R^2$, hence $r_\pi = R^2/r_0$.

[6]If $s$ and $x$ are extended outside the Lorentzian diamond using the extension of the conformal Killing vector, then the $s = 0$ slice outside the Minkowski diamond is isometric to the $s_E = \pi$ slice of the Euclidean diamond via the identification of points with the same $x$ value.

[7]One could instead generalize the definition of partition function to allow for time dependence of the boundary conditions, namely those induced on a boundary following orbits of the conformal Killing field of the Euclidean diamond. Then a saddle would exist without taking the infinite temperature limit, but it would not be a saddle of a true thermal partition function except in the infinite temperature limit.

[8]For the special case in which the energy is related to the radius by $E_{BY} = -\frac{1}{8\pi} \frac{D-2}{R} A(R)$, there does exist an exact saddle without a horizon, namely a ball in flat space.

# 3 High-temperature canonical partition function

Now consider the Euclidean gravitational path integral representation of $Z$. The integral is over $D$-metrics on a compact space with boundary geometry $S^{D-2} \times S^1$, with circumferential radii $R$ and $\beta = 1/T$, weighted by $\exp(-I)$, where $I$ is the Euclidean gravitational action, supplemented by a GHY boundary term at the system boundary.[9] Although we do not know of any exact horizon saddle in $D \geq 3$ dimensions, and strongly suspect that there is none, as explained in Section 2.1 the near-horizon limit of the Euclidean diamond can approximately meet the boundary conditions of the ensemble in the high-temperature regime $\beta \ll r_0 = R$, and a slight deformation of the near-horizon geometry can exactly meet the boundary conditions while serving as an "approximate saddle" as explained in the following subsection. In fact, it is precisely in this regime that one recovers something that matches the calculation of Ref. [30]. The conceptual difference is that the boundary around the horizon is the York boundary defining the ensemble rather than an ad hoc excision boundary, and the GHY boundary term is required in the action for the quasilocal gravitational system with that boundary, rather than being introduced by hand. The system consists of the region inside this boundary, so the extrinsic curvature in the boundary term is defined with respect to the outward normal, away from the horizon. In the following two subsections, we show explicitly how this works.

The situation is qualitatively different if there is a positive cosmological constant $\Lambda$, since then the canonical partition function does admit an exact horizon saddle. That thermal partition function has previously been analyzed in $D \geq 3$ dimensions [13,16,18,20], where the field equations admit spherical solutions containing a free parameter corresponding to the mass parameter of a Schwarzschild-de Sitter solution. It is also interesting to examine the case of JT gravity, in $D = 2$ spacetime dimensions. It turns out that, unlike in higher dimensions, there is an exact horizon saddle for $\Lambda = 0$, as well as for $\Lambda > 0$ [17,21], despite the absence of an analog of the Schwarzschild-de Sitter mass parameter. No free parameter is needed, since in JT gravity the value of the dilaton is decoupled from the boundary temperature. In view of the current interest in JT gravity, we expand on that example in some detail here.

## 3.1 $D \geq 3$: Einstein gravity

The approximate saddle we consider is the geometry $S^{D-2} \times D_\epsilon^2$, where the first factor is a sphere of radius $r_0$, and $D_\epsilon^2$ is a flat two-disk of radius $\epsilon$, with metric

$$d\ell^2 = d\rho^2 + \rho^2 d\theta^2 + r_0^2 d\Omega_{D-2}^2. \tag{3}$$

The Euclidean time is denoted by $\theta$ and there is a Euclidean horizon located at $\rho = 0$. This exactly meets the boundary conditions for a canonical ensemble with a spatial boundary geometry $S^{(D-2)}$ of radius $r_0$ and with inverse proper temperature $2\pi\epsilon$ at the boundary where $\rho = \epsilon$. However, it is not Ricci flat, since the $\rho$-$\theta$ subspace is flat while the Ricci tensor of the $(D-2)$-sphere is proportional to $r_0^{-2}$ times the sphere metric. It is thus not a bona fide saddle, but it can serve as an approximate one. The action to be computed is the bulk Einstein-Hilbert action together with the GHY boundary term. The bulk action (divided by $\hbar$) is proportional to minus the disk area $\pi\epsilon^2$ times the sphere area $A$ times the curvature scale $r_0^{-2}$ divided by a power of the Planck length (which has been set to unity),

$$I_{\text{EH}} \propto -\left(\frac{\epsilon}{r_0}\right)^2 A. \tag{4}$$

---

[9]As discussed in Ref. [19], the correct path integral is over a particular contour through the space of complex metrics, and the working hypothesis is that the contour can be deformed to pass through a saddle that is a solution to the Euclidean field equations.

The GHY boundary term is

$$I_{\text{GHY}} = -\frac{1}{8\pi} \int_{S^{D-2} \times \partial D_\epsilon^2} \mathrm{d}\theta \, \mathrm{d}\Omega_{D-2} \sqrt{\gamma} K \,. \tag{5}$$

Here $\sqrt{\gamma} = \rho r_0^{D-2}$ is the square root of the determinant of the induced metric on constant $\rho$ slices, and $K = 1/\rho$ is the trace of the extrinsic curvature of those slices. The product $\sqrt{\gamma} K$ is independent of $\rho$, so that the integral does not depend upon $\epsilon$ and evaluates to

$$I_{\text{GHY}} = -A/4 \,. \tag{6}$$

For $\epsilon \ll r_0$ the bulk action becomes negligible compared to the boundary term. It is thus plausible that the path integral is indeed dominated by the (enormous) non-contribution of the approximate saddle, and not by the (exact) flat space saddle, despite its violation of the field equations. In that sense, we may conclude that at high temperature the entropy of this canonical ensemble becomes the Bekenstein-Hawking entropy of the horizon whose area matches that of the boundary.

We emphasize that while this computation is essentially identical to that in BDF,[10] the interpretation is quite different. While they computed the action of a particular Euclidean diamond with the horizon excised, the role of the boundary here and the orientation of the boundary term derives from specifying the canonical ensemble whose partition function is being computed. The approximate saddle for this partition function can be identified with the near-horizon region of the Euclidean diamond. The entropy $A/4$ must be interpreted as associated with the plurality of near-horizon states in the ensemble, since the system consists of only the small volume between the boundary and the horizon.

## 3.2   $D = 2$: Jackiw-Teitelboim gravity

In this section we consider the canonical ensemble in a two-dimensional gravity theory, Jackiw-Teitelboim (JT) gravity [39, 40], whose field content consists of a metric tensor $g$ and a scalar field $\phi$ (a.k.a. the dilaton) non-minimally coupled to the metric. York boundary ensembles were first considered for 2D dilaton gravity black holes in Refs. [41, 42], and with a positive cosmological constant $\Lambda$ they have recently been considered in [17, 21]. Here we wish to highlight how this differs from the $D \geq 3$ case. While an infinite temperature, near-horizon limit can again be considered, it is actually not the only option since, unlike for Einstein gravity in $D \geq 3$, an exact JT saddle exists at finite temperature for both $\Lambda = 0$ and $\Lambda > 0$. We begin by introducing JT gravity and the canonical ensemble in that theory, and then consider the $\Lambda = 0$ and $\Lambda > 0$ cases in turn.

The bulk action in Euclidean signature is given by

$$I_{\text{bulk}}^{\text{JT}} = -\frac{1}{16\pi} \int \mathrm{d}^2 x \sqrt{g} \left[ \phi_0 R + \phi(R - 2\Lambda) \right] , \tag{7}$$

where $\phi_0$ is a constant and the GHY boundary action is

$$I_{\text{GHY}}^{\text{JT}} = -\frac{1}{8\pi} \int \mathrm{d}l \, K (\phi_0 + \phi) , \tag{8}$$

where $dl$ is the boundary length element. The $\phi_0$ terms of the bulk and boundary action together form the Euler characteristic of the manifold according to the Gauss-Bonnet theorem,

---

[10]BDF calculated the GHY term of Einstein gravity for causal diamonds in several different spacetimes, finding in all cases that it is equal to minus the entropy $A/4$. Since this is evaluated in the limit that the boundary approaches the horizon, all of these cases can be covered at once using the universal near-horizon geometry, which agrees with that of the approximate saddle we have considered.

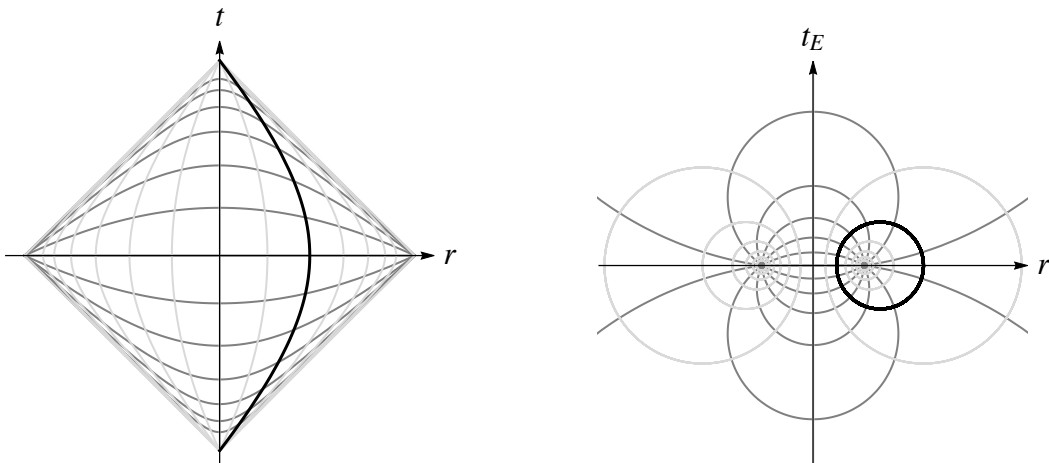

Figure 2: Lorentzian causal diamond with a $S^0$ edge (left) and its Euclidean continuation (right) in two-dimensional flat spacetime. There are two fixed points of the analytically continued conformal Killing vector corresponding to the Euclidean horizons. A York boundary along a conformal Killing orbit is drawn in black surrounding one of the horizons.

hence they are purely topological. The field equations are satisfied if and only if $R = 2\Lambda$, $\nabla^2\phi = -2\Lambda\phi$ and $\nabla_\mu\xi_\nu + \nabla_\nu\xi_\mu = 0$, where $\xi^\mu := \epsilon^{\mu\nu}\partial_\nu\phi$ (see e.g. [43]). The last condition implies that $\xi^\mu$ is a Killing vector. A canonical ensemble is defined in JT gravity by specifying the value of the dilaton $\phi_B$ and the temperature $T$ at a York boundary. In a stationary ensemble the dilaton should be constant at the boundary, hence a saddle approximating the partition function should be a solution of JT gravity that contains a circle of circumference $\beta = 1/T$ on which the value of the dilaton matches $\phi_B$.

For the case $\Lambda = 0$, the general solution is a flat metric, $dx^2 + dy^2$, and a linear dilaton, $\phi = ax + by + c$, where $a, b, c$ are constants. For $a = b = 0$ and $c = \phi_B$ there is an exact saddle solution that meets the boundary condition of a constant dilaton on a thermal boundary circle. The bulk action of this solution vanishes because the Ricci scalar is zero and the boundary term evaluates to [44]

$$I_{\text{GHY}}^{\text{JT}} = -\frac{1}{4}(\phi_0 + \phi_B). \tag{9}$$

Thus, for any temperature and constant dilaton the action of the saddle is given by minus the JT horizon entropy, which can be interpreted as the entropy of a causal diamond in flat space. Note the horizon entropy is independent of the size of the diamond. On the right in Figure 2 we show the analytic continuation of the conformal Killing orbits in Euclidean flat space. Since the edge $S^0$ of the diamond consists of two disconnected points, there are two fixed points of the conformal Killing field, i.e., two Euclidean horizons.

Next we consider the case of a positive cosmological constant, for which the metric is that of Euclidean de Sitter space, i.e., a 2-sphere, with radius $L = 1/\sqrt{\Lambda}$. The Killing vectors of the 2-sphere generate rotations about an axis. If we orient the coordinates so that the Killing vector determined by $\phi$ corresponds to the azimuthal vector field, $\partial_\varphi$, the condition that $\xi^\mu$ be a Killing vector implies that $\phi \propto \cos\theta$. Since this is the $\ell = 1$, $m = 0$ spherical harmonic, we also have that $\nabla^2\phi = -(2/L^2)\phi$, so this dilaton field satisfies in addition the required elliptic equation. The solution to the field equations is therefore unique up to the orientation of the dilaton Killing axis and the multiple of $\cos\theta$. We note that the dilaton is globally regular over the 2-sphere, although we shall need it only within a patch.

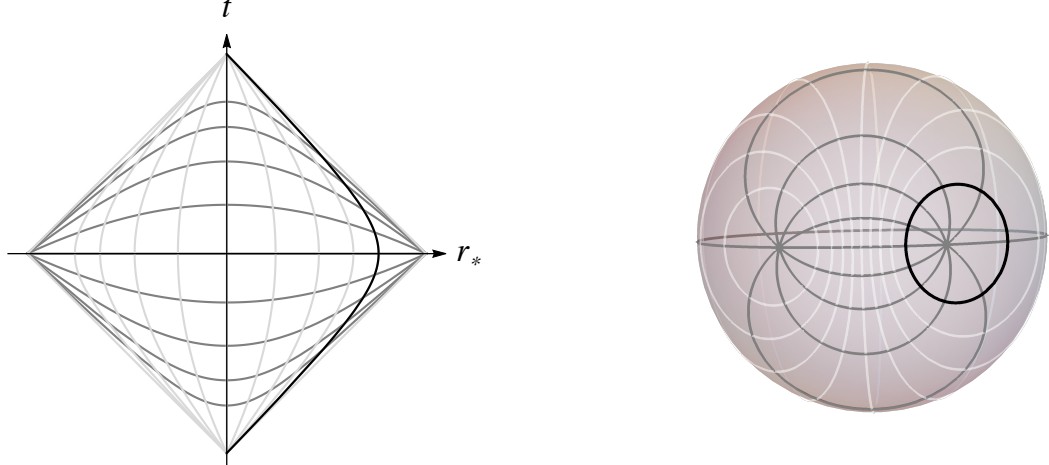

Figure 3: Lorentzian causal diamond with a $S^0$ edge (left) and its Euclidean continuation (right) in two-dimensional de Sitter spacetime. On the left $t$ is the Killing time of the de Sitter static patch and $r_*$ is the tortoise coordinate. The Euclidean diamond covers the entire two-dimensional Euclidean de Sitter manifold, $S^2$. A conformally stationary York boundary is drawn in black.

As is well-known, the entire 2-sphere is the analytic continuation of a static patch of Lorentzian de Sitter space, continuing the Schwarzschild-like Killing time coordinate $t$ to imaginary values, and the analytic continuation of the static patch Killing vector $\partial_t$ is a rotation Killing vector of the sphere. Perhaps surprisingly, if we analytically continue the *conformal* Killing time in a causal diamond smaller than a static patch in de Sitter spacetime, the resulting space still corresponds to the entire Euclidean 2-sphere [17, 30]. To see into how this works geometrically, note that the analytically continued curvature must again be constant, so it forms some part of the 2-sphere, and the analytic continuation of the conformal Killing vector is a conformal Killing vector of the sphere. We again arrive at the *entire* 2-sphere, because the static patch and a smaller causal diamond are related by a conformal isometry, so their analytic continuations must also be so related. Another way to see this is that the points along the right side of the diamond lie at zero Lorentzian distance from each other, so they map to a single point in the Euclidean space, and similarly for the points on the left side. The top and bottom vertices of the Lorentzian diamond lie at positive and negative infinite conformal Killing time, which in the Euclidean continuation means that they are identified.

The conformal Killing orbits of the Lorentzian diamond have constant acceleration [29], so their analytic continuations are circles on the sphere. The pattern of conformal Killing orbits (and the orthogonal curves) can in principle be found by placing circles with the appropriate acceleration (i.e., extrinsic curvature on the sphere) at the corresponding points of the maximal slice of the causal diamond, which is shared with the analytic continuation. Alternatively, they can be obtained by applying a suitable conformal transformation to the orbits of a rotational Killing field on the sphere. In fact, that is how we generated the plot on the right in Figure 3. Conveniently, the conformal group of the sphere is $SL(2, \mathbb{C})$, the double cover of the Lorentz group, and its action on the sphere can be realized by its action on the null rays on a light cone in Minkowski spacetime, when identifying the sphere with a cut of the light cone at a fixed Minkowski time in some frame. (This is the famous fact that Lorentz transformations conformally transform the celestial sphere.) The figure is generated by a boost (with rapidity 1.5) perpendicular to the rotation axis of the rotational Killing vector. The relativistic "beaming" effect causes the two poles to approach each other.

If we begin with the aim of computing the entropy of a causal diamond as viewed by an observer inside, it would seem natural to identify the boundary in the saddle with an orbit of a conformal Killing field, like the black circle in Figure 3, since that is the analytic continuation of a constant acceleration orbit in the Lorentzian diamond. Concerning the boundary value of the dilaton, there is a solution whose pole agrees with the fixed point of the conformal Killing flow, but unlike the constant dilaton in flat space this solution is not constant on the black circle, since the fixed point does not lie at the center of that circle, so this solution cannot serve as a saddle for the canonical partition function.

We hence consider instead a canonical ensemble with a prescribed constant value of the dilaton on the boundary and a given temperature, and look for a saddle that meets those boundary conditions. Unlike in Einstein gravity in the higher dimensional case, where a saddle arises only approximately in the high-temperature regime, in JT gravity there is an exact saddle. That solution has a pole at the center of the boundary circle, rather than at the fixed point of the conformal Killing flow. That is, the saddle is invariant under the rotational Killing flow that preserves the boundary. The portion of the 2-sphere between the York boundary and the pole, together with the dilaton solution $\phi_H \cos\theta$, form an exact saddle of the ensemble that meets the boundary conditions with boundary at $0 < \theta_B = \sin^{-1}(\beta/2\pi L) < \pi/2$, and horizon value of the dilaton $\phi_H = \phi_B / \cos\theta_B$. Note that since this canonical ensemble is defined without reference to the conformal Killing vector of the causal diamond initially under consideration, its saddle has no relation to that diamond. Instead, it corresponds to a portion of the Euclidean static patch.

In fact, there are two configurations consistent with the boundary conditions, since there are two poles concentric with a given circle on the 2-sphere. In the Lorentzian counterpart, i.e. the de Sitter static patch, if they arise from a dimensional spherical reduction of the near-Nariai black hole geometry the two poles can be interpreted as the "cosmological" ($\phi_H/\phi_B > 0$) and "black hole" ($\phi_H/\phi_B < 0$) horizon. The on-shell action for these configurations was computed in [17, 21] and is given by

$$I_{\text{tot}}^{\text{JT}} = I_{\text{bulk}}^{\text{JT}} + I_{\text{GHY}}^{\text{JT}} = \beta E_{\text{BY}} - S, \tag{10}$$

where $\beta$ is the proper inverse boundary temperature, and the Brown-York energy $E_{\text{BY}}$ and horizon entropy $S$ are, respectively,

$$E_{\text{BY}} = \pm \frac{\phi_B \beta/2\pi L}{8\pi\sqrt{1-(\beta/2\pi L)^2}}, \qquad S = \frac{\phi_0}{4} \pm \frac{\phi_B/4}{\sqrt{1-(\beta/2\pi L)^2}} \equiv \frac{\phi_0 \pm \phi_H}{4}. \tag{11}$$

The plus sign corresponds to the cosmological configuration, whereas the minus sign is associated to the black hole configuration. In the last term $\phi_H$ is the value of the dilaton at the cosmological horizon, and $-\phi_H$ is the value at the black hole horizon. In the infinite temperature limit, $\beta \to 0$, the boundary circle shrinks to zero size, so $\phi_H \to \phi_B$. In the limit $\beta \to 2\pi L$, the temperature drops to the de Sitter temperature, and a regular saddle exists only for $\phi_B \to 0$. No saddle of temperature lower than $1/2\pi L$ exists. The free energy ($F = I_{\text{tot}}^{\text{JT}}/\beta$) of the cosmological configuration is always lower than that of the black hole configuration [17, 21], hence the cosmological configuration dominates the ensemble.

# 4 Discussion

We set out to clarify the physical principles underlying the derivation of BDF [30] of the entropy associated with causal diamonds. In all cases it was found there that a boundary term localized at the horizon yields the familiar Bekenstein-Hawking area entropy when the action is evaluated on a Euclidean analytic continuation of the corresponding Lorentzian diamond.

While we do not doubt the correctness of the answer, what seemed less clear to us is the question.

We took the viewpoint that to identify an entropy we should first specify the gravitational ensemble. To this end we adopted York's method of introducing a system boundary at which the ensemble parameters are specified, and looked for a saddle point approximation to the partition function for the canonical ensemble. For a finite boundary at a generic temperature, if the cosmological constant vanishes the only saddle for Einstein gravity in $D \geq 3$ spacetime dimensions corresponds to "hot flat space", which has vanishing entropy. However, the calculation of BDF can be recovered if one considers the regime in which the ensemble temperature becomes very large, since then an approximate saddle with a horizon emerges, which corresponds to the near horizon geometry. (This ensemble interpretation of the calculation also justifies the contibution of the horizon boundary term as well as its orientation, which were not clear to us in the framework of the BDF calculation.) One can thus interpret the entropy as being associated with the near horizon states, and its universal form (1/4 per Planck area) is independent of the particulars of the causal diamond. By contrast, there does exist an exact saddle with a horizon in $D = 2$ for JT gravity, with the dilaton constant everywhere in spacetime. The action of this saddle gives minus the horizon entropy of a diamond for any temperature.

We also noted that when there is a cosmological constant, the partition function *does* admit a horizon saddle for a wide range of finite temperatures. This was already found earlier in Einstein gravity in $D \geq 3$ spacetime dimensions [13,16,18,20], as well as in JT gravity in $D = 2$ dimensions [17,21]. In this case, the entropy is again given (at leading order) by the area law, or its JT gravity generalization, even though the saddle includes a finite volume between the York boundary and the horizon. However, the saddle in these cases does not correspond to the original causal diamond one set out to associate an entropy with. In Einstein gravity, for example, the saddle is a Schwarzschild-de Sitter geometry with a value of the mass parameter that is selected by the ensemble parameters, and is never just an empty causal diamond that is smaller than the de Sitter static patch.

Our conclusion is that the question for which the calculation of BDF is the answer is "what is the entropy of an infinite temperature canonical ensemble defined by a boundary localized just outside the saddle horizon"? In a sense, this is satisfactory since, as pointed out by BDF, the near-horizon state has long been understood to behave, at least in a semiclassical approximation, as a thermal state with diverging proper temperature as the horizon is approached by a static observer. On the other hand, the presence of the ensemble boundary just outside the horizon is a physical ingredient alien to the original question intended by BDF, namely, what is the entropy—or the dimension of the Hilbert space—associated with a causal diamond on its own, without an artificial boundary inserted? To address this latter question one should consider the limit in which the boundary shrinks to zero size and disappears. It was explained in [19] (following the original suggestions of [45, 46]) how this leads to the entropy of an "empty" de Sitter static patch when a positive cosmological constant is present. This amounts to an interpretation of the original Gibbons-Hawking sphere partition function [9] as a computation of the trace of the identity operator on the Hilbert space, i.e., the dimension of the space of states. When the cosmological constant vanishes, or for a diamond smaller than the de Sitter static patch, this calculation fails, since no semi-classical saddle exists. In another paper [47] we have taken up the challenge to make sense of the entropy of those causal diamonds.

## Acknowledgments

We thank Batoul Banihashemi, Patrick Draper and Andrew Svesko for valuable discussions.

**Funding information** The research of TJ is supported in part by National Science Foundation grant PHY-2012139. This work was done while TJ was a Visiting Fellow Commoner at Trinity College, Cambridge. He is grateful to the college for hospitality and support. MRV is supported by SNF Postdoc Mobility grant P500PT-206877 "Semi-classical thermodynamics of black holes and the information paradox".

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
