# Peer review of "Entropy of causal diamond ensembles"

_SciPost Physics, doi:SciPost Phys. 15, 023 (2023)_

## Round 1 · Referee Report · Anonymous (Referee 3) · 2023-4-8

Report

The paper attempts to formulate the thermodynamic properties of the static patch more clearly by introducing an auxiliary boundary, of the type considered in the early work of York. The methodology of Euclidean gravity is employed, and various spacetime dimensions as well as vanishing and non-vanishing cosmological constants are considered.

Although the problem is considered at the level of the semiclassical saddle, which only exists for $\Lambda>0$, it seems that Dirichlet boundary conditions are chosen at the boundary to formulate the problem, and it would be good to state that more explicitly to contrast other boundary conditions considered for such auxiliary boundaries (e.g. conformal boundary conditions discussed in 0704.3373, 1805.11559, 2103.15673, 1110.3792 among other places). I encourage the authors to expand on this but leave it as an option.

The paper nicely expresses salient differences between $\Lambda=0$ and $\Lambda>0$ and contributes to the general question of horizon thermodynamics. I support the publication.

  • validity: -
  • significance: -
  • originality: -
  • clarity: -
  • formatting: -
  • grammar: -

Author:  Manus Visser  on 2023-04-23  [id 3609]

(in reply to Report 1 on 2023-04-08)

We thank the referee for the suggestion that we expand on the issue of boundary conditions. The issue of boundary conditions is an important one. For the present paper we feel it is sufficient to emphasize that the Dirichlet choice has been made, that this choice might not be dynamically viable, that we expect the results will likely not depend on that choice, and that it would be interesting to investigate that question. To this end, we have replaced the opening of section 2.1 with text that addresses these points.

---

## Round 2 · List of Changes

We have replaced the opening of section 2.1 with text that addresses the issue of boundary conditions.

---

## Editorial Decision

published